# Improving a Street-Based Geocoding Algorithm Using Machine Learning Techniques

**Kangjae Lee [1] , Alexis Richard C. Claridades [1,2] and Jiyeong Lee [1,*]**

[1] Department of Geoinformatics, University of Seoul, 163 Seoulsiripdae-ro, Dongdaemun-gu, Seoul 02504, Korea; kkooring@uos.ac.kr (K.L.); uosgrad2019012@uos.ac.kr (A.R.C.C.)

[2] Department of Geodetic Engineering, University of the Philippines Diliman, Quezon City 1101, Philippines

* Correspondence: jlee@uos.ac.kr

**Abstract:** Address matching is a crucial step in geocoding; however, this step forms a bottleneck for geocoding accuracy, as precise input is the biggest challenge for establishing perfect matches. Matches still have to be established despite the inevitability of incorrect address inputs such as misspellings, abbreviations, informal and non-standard names, slangs, or coded terms. Thus, this study suggests an address geocoding system using machine learning to enhance the address matching implemented on street-based addresses. Three different kinds of machine learning methods are tested to find the best method showing the highest accuracy. The performance of address matching using machine learning models is compared to multiple text similarity metrics, which are generally used for the word matching. It was proved that extreme gradient boosting with the optimal hyper-parameters was the best machine learning method with the highest accuracy in the address matching process, and the accuracy of extreme gradient boosting outperformed similarity metrics when using training data or input data. The address matching process using machine learning achieved high accuracy and can be applied to any geocoding systems to precisely convert addresses into geographic coordinates for various research and applications, including car navigation.

**Keywords:** geocoding; machine learning; address; alias

## 1. Introduction

Addresses are one of several methods that people perceive location as a textual natural language description. Geographic locations are associated with 80% of the information local government use, and addresses are related to most of the geographic locations [1]. Especially in urban areas, they are used to communicate and reference a spatial location through direct and indirect methods [2,3]. Since these addresses serve as a link to locate demographic, social, economic, or environmental attributes, Geographic Information System (GIS) proves to be a useful tool across application domains. Integration allows for further analysis and exploring relationships in the context of location, even up to the individual level [4–7].

Addresses are critical components of geocoding. Geocoding is a process to transform the addresses into geographic coordinates and has become increasingly important in many fields to locate the addresses on a map in GIS. The geocoding was initially developed by the U.S. Bureau of the Census to "allocate population accurately within blocks, census tracts, and other geographic areas, without the expense of dispatching enumerators to every dwelling unit in the country" [8]. While Global Positioning System (GPS) may be desirable due to its accuracy, geocoding is regarded as more reliable since GPS may not always be feasible or affordable [9]. Additionally, geocoding supports an expanded view of addresses, including not just structured hierarchical definitions of locations, but also building names, postal codes, and telephone area codes [3]. These days, geocoding has been widely used

in many different fields and research areas, including transportation, public health, and logistics, for mapping not only a particular location itself, but also events and phenomena [10–12].

Address matching is an essential process in geocoding [13], where input addresses are matched with the counterparts in a reference database to get corresponding coordinates eventually from the matched addresses. This step, however, forms a bottleneck for geocoding accuracy, as precise input is the biggest challenge for establishing perfect matches [14]. Matches still have to be established despite the inevitability of incorrect address inputs [15] such as misspellings, abbreviations, informal and non-standard names, slangs, or coded terms. For example, in the street-addressing system, a street name 'Park Avenue' can have different aliases, such as 'Park Ave.', 'Park Av.'. It can have variants input by different users, and the geocoding system needs to match these aliases with the corresponding counterparts in a reference database. Regarding geocoding, previous studies widely used similarity metrics for matching addresses [16,17]. Each similarity metric, however, has its characteristic in matching addresses showing different performance in different cases, and we need to find a way to combine such different characteristics for ideally matching addresses correctly.

Thus, this study suggests an address geocoding algorithm using machine learning techniques to enhance the address matching implemented on street-based addresses. The developed address geocoding algorithm contains three modules—address parsing, address matching, and address locating. A regex-based parsing method divides the input addresses into elements of the street-based system. For the address matching, this study introduces a way to combine similarity metrics using machine learning to improve performance. We evaluate the suggested address matching, especially the performance of three machine learning models in the address matching process by comparing to multiple text similarity metrics, which are present in previous studies for text matching [18]. This study contributes to the enhancement of geocoding by adopting machine learning techniques for address matching and will be helpful for relevant geocoding research and applications.

The sections in this paper are structured as follows: Section 2 describes past studies on geocoding. Section 3 demonstrates the components of the suggested geocoding algorithm. Section 4 deals with the evaluation of the developed geocoding algorithm, and the findings in this study are discussed and concluded in Section 5.

## 2. Overview of Geocoding and Its Advancement

In this section, we will review the general steps involved in geocoding and existing studies on geocoding and word matching.

### 2.1. General Steps for Geocoding

As one of the primary functions of a GIS [19], geocoding works to assign positions, such as latitude and longitude, into textual addresses using a reference database [3,5,10,15,19]. These addresses are attached to datasets that are applicable across various applications domains; hence by geocoding, a bridge between spatial and attribute data is established [20]. This association to the geographic information enables not just visual display through accurate maps, but also the application of more in-depth spatial analysis [21,22]. This integration strengthens the function of GIS as a vital tool across various fields of interest such as urban planning and management [13], human activity and movement studies [5], health [9,23,24], emergency dispatching [7], traffic accidents [12], and management of administrative data [4].

Generally, the geocoding process includes three steps—parsing, matching, and locating, as shown in Figure 1 [5]. Previous studies have presented analogous procedures that mostly aim to accept textual input, perform a preprocessing step to this input, and perform a match to a database to return coordinate value pairs for position [8,22,25]. Parsing converts unstructured or semi-structured input addresses into structured ones. This process is crucial in giving out precise location, even if input addresses or even the address databases are imprecise and vague [3]. Input addresses may have problems matching due to their unstructured forms. Thus it is essential to capture meaningful

units of addresses and enhance the quality of these address elements, which is critical in improving geocoding accuracy [22]. Matching compares and links the structured input addresses to an address reference database. The address reference database includes the information on the addressing system to be matched with the input addresses. Locating finds coordinates based on the results from the matching process.

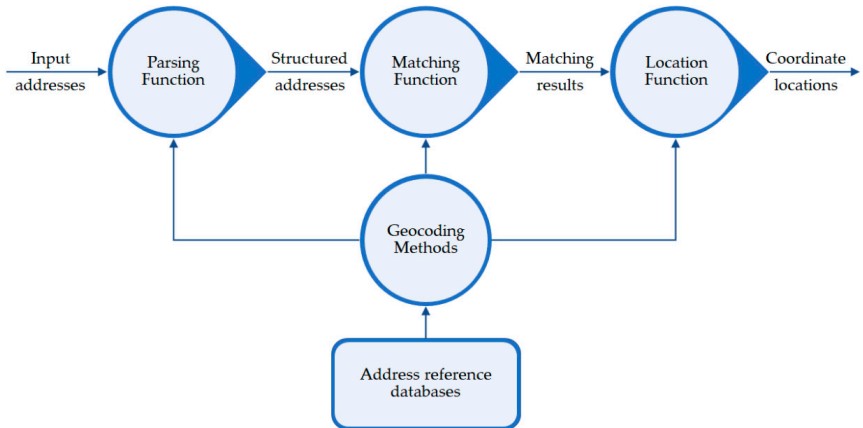

**Figure 1.** General steps for geocoding.

### 2.2. Existing Studies on the Advancement of Geocoding and Word Matching Using Machine Learning

Geocoding has been the subject of multiple studies, particularly on the improvement of its process. Lee (2009) [5] focused on an area-based address geocoding method since street-based address geocoding methods are limited to Western countries and thus, do not apply to countries with area-based addressing systems. The suggested area-based address geocoding method was able to define house locations within a block accurately. In order to do that, this method models the boundaries with block number and building number range information, and the network distance defines the geographic coordinates of houses along segments of block polygons using a linear interpolation technique. Furthermore, a 3D address geocoding method was also proposed based on the 3D indoor geocoding [26]. The 3D address includes an address for building and an indoor address, such as an apartment number. Especially for the 3D indoor geocoding, network models of buildings are ideal. The constructed network model forms the foundation of calculating network distances from an interpolation method and finding location information in a building. Lee et al. (2017) [27] also suggested a novel idea for 3D indoor geocoding that utilizes optical character recognition to detect the current indoor location and semantic queries to determine a destination of interest. Yao et al. (2015) [6] suggested three fuzzy matching algorithms for Chinese address matching based on a full-text search. They focused more on user input and result control rather than address standardizations and models, which current researchers pay attention to. With the three fuzzy matching methods, the address matching engine was able to achieve higher match efficiency than the traditional database retrieval. In addition, it guaranteed greater freedom on user input and result control and showed very high accuracy (100%).

Matci and Avdan (2018) [14] proposed a method to standardize addresses to improve geocoding results. The address data undergo parsing, semantic analysis, and reformatting through the natural language process. The developed method was tested on 233 primary school addresses using Google Geocoding API and ArcGIS geocoding API. In addition, the test data were standardized in three formats—the Turkish National Post Telephone Telegraph, Google, and ArcGIS. The results indicated that the standardized addresses significantly improved the accuracy of the geocoding results. Especially, when the addresses were standardized in the Google format and geocoded using Google Geocoding API, it showed the highest accuracy (99.1%).

Regarding word matching, some studies suggested ways of using machine learning methods [13,18,28,29]. Christen et al. (2006) [28] developed a novel geocode match engine with

a rule-based approach to find an exact match or other approximate matches. Especially, hidden Markov models were adopted to achieve better address standardization accuracy. The developed geocoding engine achieved 94.94% matches at different levels (address, street, locality). Choi and Lee (2019) [18] proposed an approach to the alias database management for efficient POI (Point of Interest) retrieval. The authors adopted Word2vec, a simple neural network structure, for word embedding to convert text data into the form of numeric vectors. The word embedding is capable of making a machine learning model understand similar meanings of words. The suggested method determines the match of a given POI name and the corresponding POI in the alias DB based on text similarity. The most similar word with the similarity degree of 60% or more is retrieved, and the user confirms whether it is the correct one. Santos et al. (2018) [29] used supervised machine learning methods for combining multiple similarity metrics concerning toponym matching. The use of machine learning with multiple similarity metrics has the benefits of avoiding setting similarity thresholds manually. The authors showed that the methods based on machine learning outperform the individual usage of similarity metrics with setting a manual threshold. Lin et al. (2020) [13] introduced a novel address matching method based on deep learning techniques for identifying the semantic similarity between address records. The suggested method computed the semantic similarity of the compared address records for determining whether they match. It was evaluated using the Shenzhen Address Database and achieved 97% of accuracy outperforming other current address matching methods.

This study proposes an algorithm for accurate and efficient geocoding. For the matching process, machine learning methods with multiple similarity metrics are used, like Santos et al. (2018) [29]. Compared to Santos et al. (2018) [29], this study introduces more similarity metrics for better performance of matching. Moreover, the proposed method utilizes a hyper-parameter tuning of machine learning methods to select the best machine learning method for the matching process. An alias DB is the source of training data for machine learning methods. Regarding the quality of the training data, we also apply different ratios of matching and non-matching pairs to understand the effect of different combinations of the pairs on the performance of matching.

## 3. Geocoding Algorithm Using Machine Learning Techniques

In this section, we discuss the proposed geocoding algorithm divided into three major parts—address parsing, address matching using machine learning, and address locating. The input addresses undergo a parsing process in order to determine which parts of the text belong to each address component. Then, each of these components undergoes a matching process integrated with a machine learning method in order to increase matching accuracy. We then compare resulting matches to a reference database for calculating coordinate information. Figure 2 summarizes this process, and succeeding sub-sections will discuss these in detail.

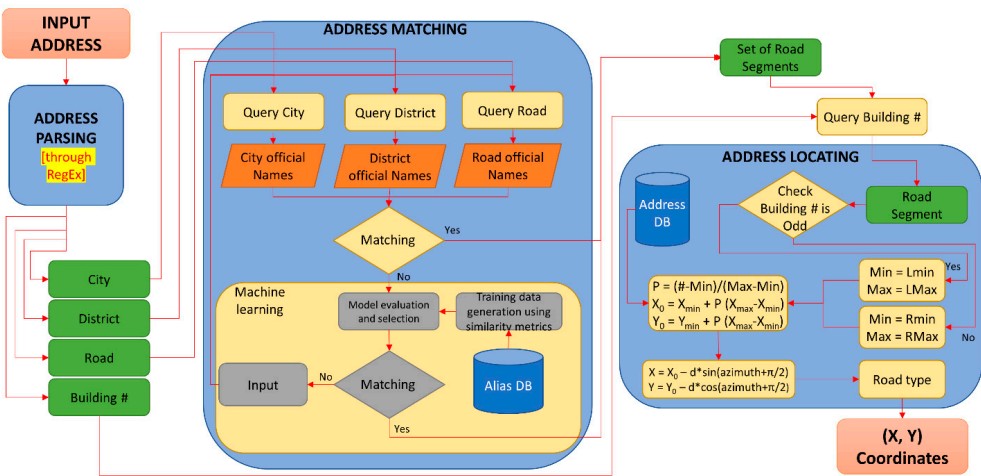

**Figure 2.** Methodology for implementing the geocoding system using machine learning.

*3.1. Address Parsing*

Parsing is the analysis of a sequence of characters and, in particular, breaks given texts into meaningful pieces. In this paper, we used Korean addresses, based on the road name address system implemented in 2014. This system follows a hierarchy of administrative units, optionally beginning with a province name (suffixed by "do"), city ("si"), district ("gu"), that precedes the road name (ending with "daero", "ro", or "gil", depending on the number of lanes) and building number [30]. In the developed algorithm, the parsing (left side of Figure 2 and Algorithm 1) divides the input address into the city, district, road, and building number so that each piece can work adequately in the matching or locating process. Then, the building number goes through the query building number to decide the road segment used to compute the geocoded location. The rest of parsed city, ward, and road goes to the matching component and we match them with the records in the Alias DB.

Regular expressions, or simply Regex, is a programming tool used in many languages such as Python, Java, Perl, and PHP [31]. Regexes allow expression of patterns and repetitions in texts [32], so they perform well in detecting parts of an address input. In the case of Korean addresses, they may be used to identify address elements more flexibly, compared to purely whitespaces as delimiters (since each element may contain a whitespace in between). Based on the character suffixes used in Korean Addresses, we constructed a regex to detect characters or numerals for each element from the raw input of the user.

---

**Algorithm 1** Parsing process

---

1: **Input**: *AddressText*
2: ***Output***: *City, District, Road, Building Number*
3: **Algorithm**
4: Initialize the parameters of the *City, District, Road, Building Number, keys*, and *match*
5: **define** *regex_dictionary* keys:
6:　　*city = [province suffix][space] <alphanum.city>[city suffix]*
7:　　*district = [city suffix][space] <alphanum.district>[district suffix]*
8:　　*road = [district suffix][space] <alphanum.road>[maj. rd. suffix][num][min. rd. suffix]*
9:　　*bldgno = [any alphanum][space] <num.bldgno>[whitespace]*
10: **for** every key in *regex_dictionary*:
11:　　*match* = Search (*AddressText*)
12: **for** every *match*:
13:　　**if** *key == 'city'*:
14:　　　　*City = match*
15:　　**else if** *key == 'district'*:
16:　　　　*District = match*
17:　　**else if** *key == 'road'*:
18:　　　　*Road = match*
19:　　**else if** *key == 'bldgno'*:
20:　　　　*BldgNo = match*

---

*3.2. Address Matching Using Machine Learning*

The parsed city, district, and road names undergo the address matching using machine learning, resulting in a set of road segments through the matching process. The address matching process helps to select a set of road segments to be sent to the address locating process. The city, district, and road names are matched one or two times in the matching process to find the records in the Alias DB for geocoding correctly (Algorithm 2 and in the middle of Figure 2).

---

**Algorithm 2** Address matching process

---

1: **Input**: *City, District, Road, Traindata*
2: **Output**: *MatchedName*
3: **Algorithm**
4: Initialize the parameters of the *S = 0, SearchName, Result, SearchResult, Test, OfficialName*, and
5: *Probs*
6: *MLT* = MachineLearningTrain(*Traindata*)
7: **while** (*S* < 2)
8: 　**if** *S* = 0
9: 　　*SearchName = City, District*, or *Road*
10: 　**else if** *S* = 1
11: 　　*SearchName* = input("Input city, dictrict, or road name once again")
12: 　*Result* = MatchOfficialName(*SearchName*)
13: 　**if** length(*Result*) > 0
14: 　　*SearchResult = Result*
15: 　　*S* = 2
16: 　**else**
17: 　　**for** *i* = 0 to number of official names
18: 　　　*Test*[*i*] = GenerateSimMetrics(*SearchName, OfficialName*[*i*])
19: 　　　*Probs*[*i*] = MLT.PredictProbability(*Test*[*i*])
20: 　　**if** there is a matched official name having the highest probability in *Probs*
21: 　　　*SearchResult* = the matched official name
22: 　　**else**
23: 　　　*SearchResult* = ''
24: 　　**if** the matched result is correct
25: 　　　*S* = 2
26: 　　**else**
27: 　　　*S* = 1
28: **end while**

---

In the first matching, the city, district, and road names undergo simple matching with their official names. If there is no identical official name found for the city, district, or road name, the second matching is subsequently involved using machine learning to supplement the first simple matching. Before matching using machine learning, we generate training data from the Alias DB, and select the best machine learning model through the evaluation of 17 text similarity metrics (Table 1) and three machine learning models. Each record of Alias DB has an official name and the aliases, while the training data consist of matching pairs and non-matching pairs. The matching pairs refer to the pairs of one official name and one alias of the official name. On the other hand, the non-matching pairs ae defined by the aliases and other unmatched official names in the Alias DB. In this study, we tested different combinations of matching and non-matching pairs for the training data since the accuracy of matching results may vary depending on the ratio of matching and non-matching pairs consisting of training data. For matching and non-matching pairs, we used variables calculated using 17 text similarity metrics to train machine learning models.

Previous studies have widely used similarity metrics for word matching [16,18,33,34]. This methodology uses edit-based and token-based similarity metrics. Edit-based similarity is more applicable for short phrases and compares only the characters. Hence, it is simple, yet it is inefficient for long phrases and computationally expensive. On the other hand, token-based similarity explores text as a set of tokens (words) and is more applicable for long texts.

**Table 1.** Seventeen kinds of similarity metrics.

| Edit-Based Similarity Metric | Token-Based Similarity Metric |
| :---: | :---: |
| Jaro [35] | Cosine [36] |
| Jaro-Winkler [37] | Tversky [38] |
| Jaro-Winkler Reversed [29] | Overlap [34] |
| Jaro-Winkler Sorted [28] | Bag [39] |
| Hamming [40] | Jaccard [41] |
| Mlipns [42] | Sorensen_Dice [43] |
| Strcmp95 [37] | Monge_elkan [33] |
| Needleman-Wunsch [44] | |
| Gotoh [45] | |
| Smith_Waterman [46] | |

Three different kinds of machine learning methods—support vector machine (SVM), random forest (RF), extreme gradient boosting (XGB)—were tested to find the best method. SVM is a supervised machine learning method. The goal of SVM is to find a hyperplane in an N-dimensional space that best separates data points. A hyperplane that has the maximum margin between data points is selected as the best decision boundary to classify the data points. Given some training data D:

$$D = \{(x_i, y_i) | x_i \in R^m, \ y_i \in \{-1, 1\}\}_{i=1}^{n} \tag{1}$$

where $x_i$ is an m-dimensional real vector, $y_i$ is either 1 or −1, indicating the class of input vector $x_i$. Two parallel hyperplanes (Figure 3) are defined such that $w^T x + b = 1$ and $w^T x + b = -1$. Maximizing the distance between these two hyperplanes defines the maximum-margin hyperplane.

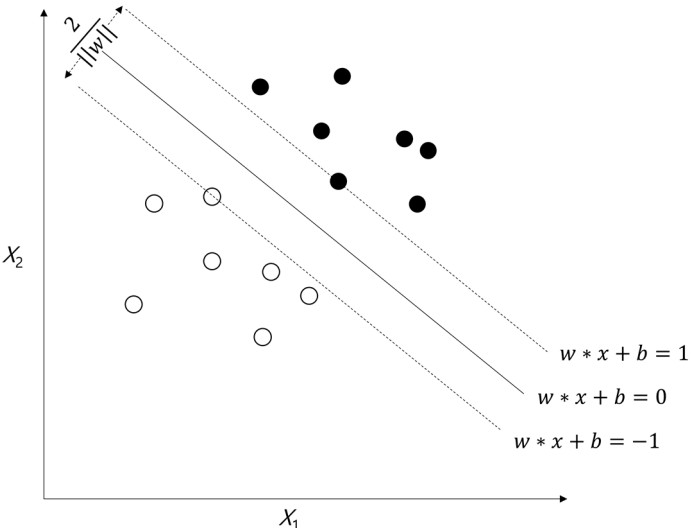

**Figure 3.** Support vector machine algorithm.

RF is also a supervised machine learning method and built upon decision trees on data points (Figure 4). It makes predictions from each of the trees with different features and selects the best solution with majority voting. By majority voting, RF can reduce the over-fitting.

XGB, as a supervised machine learning method, is an implementation of gradient boosted decision trees for more efficient computation and the increase of performance. In the gradient boosted trees, each tree in boosting is a weak learner and tries to minimize the errors of the previous tree to make a strong learner.

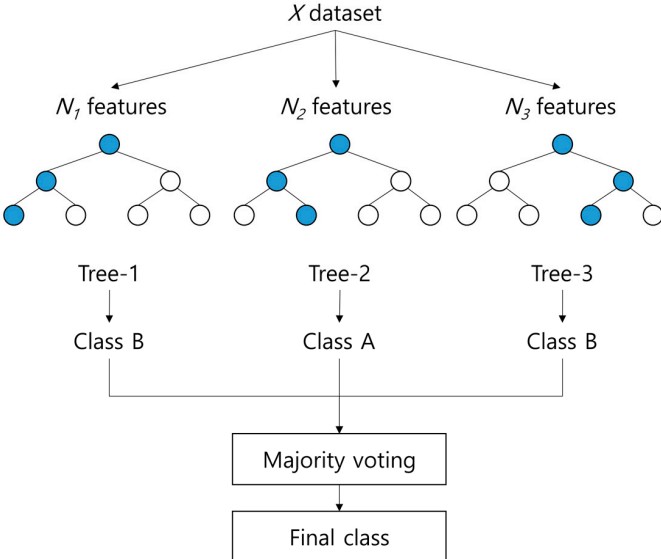

**Figure 4.** Random forest algorithm.

We performed a 10-fold cross-validation to evaluate the performance of the matching process. This method splits the entire data set into 10 sets with equal size, and selects one single set as a test set. Remaining sets become training sets, and the cross-validation is repeated 10 times by using each test set only once. The performance of three machine learning models was compared with each similarity metric to see whether the machine learning models outperform conventional methods.

To understand the minimum number of similarity metrics needed and choose the best model, we tested different numbers of similarity metrics. To do this, first, different similarity metrics were arranged in descending order by their accuracy and grouped into 2–17 based on the order. Second, the least number of similarity metrics showing higher accuracy of machine learning models than all the 17 similarity metrics was determined (Figure 10). Last, we chose the best model with the minimum number of similarity metrics that shows the highest accuracy among the three machine learning models. We measured accuracy for each similarity metric and the three different machine learning models. Especially, for each similarity metric, a threshold value needs to be set, determining whether a given pair of texts matches or not. Therefore, to find the optimal threshold, different thresholds need to be tested (Algorithm 3). This methodology tests different threshold values ranging from 0.00 to 1.00 by the increment 0.05 for each similarity metric and sets the optimal threshold that has the highest accuracy.

---

**Algorithm 3** Choosing optimal thresholds for similarity metrics

---

1: **Input**: *Traindata, Threshold*
2: **Output**: *OptimalThreshold*
3: **Algorithm**
4: Initialize the parameters of the *Predicted*, *Label*, *Acc*, *Alpha,* and *OptimalThreshold*
5: **for** $j$ = 0 to number of similarity metrics
6:    **for** $k$ = 0 to length(*Threshold*)
7:      **for** $i$ = 0 to length(*Traindata*)
8:        **if** *Traindata*[$i$][$j$] >= *Threshold*[$k$]
9:          *Predicted*[$i$] = 1
10:       **else**
11:          *Predicted*[i] = 0
12:      *Acc*[$k$] = AccuracyScore(*Label, Predicted*)
13:      *Alpha*[$k$] = *Threshold*[$k$]
14:    *MaxIndex* = Argmax(*Acc*)
15:    *OptimalThreshold*[$j$] = Alpha[*MaxIndex*]

---

When the first and second matching do not find a matched official name, the city, district, or road name is input manually and then the matching is undertaken once again. A set of road segments results from the end of the address matching process. As a result of the matching, candidates of road segments are selected and sent to the locating process with the parsed building number. The combination of road segments and the building number can help to narrow down to a final road segment necessary in locating.

### 3.3. Address Locating

The U.S. Census Bureau performs address locating with address ranges [47]. As illustrated in Figure 5, each record in the address database includes two address ranges for the left and right sides of the road. The left side has odd-numbered addresses, and the right side has even-numbered addresses. For example, geocoding the address with the building number '105' means we must first identify how this building number was assigned. In some territories, such as in the United States, the Bureau assigns this number from the address number range of the start and end nodes located at the centerline of the roads intersecting the concerned road segment. The building number arises from interpolation. Since the numbering begins at the centerlines, not within the segment itself, the widths of these perpendicular roads containing the start node, or start width, and the end node, or end width, must be taken into consideration. Therefore, to geocode this building's address, its location is approximated on the left side (odd-numbered addresses) of the road 'Park Avenue' using a linear interpolation method. In order to adjust the aforementioned road's length, the road types of the adjacent roads (i.e., road containing the start node and the one containing the end node) are taken into consideration. Figure 5 (Left) illustrates this case.

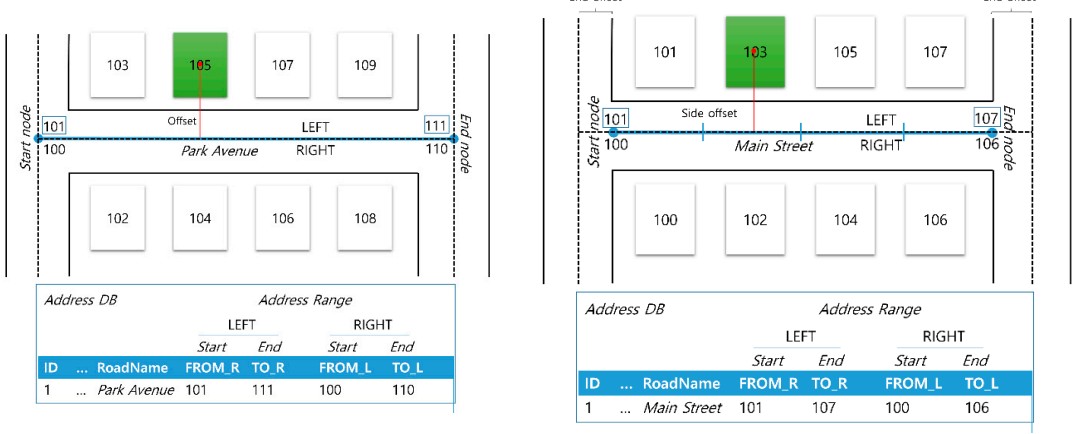

**Figure 5.** Address locating that assigns building numbers proportionally along roads (**Left**) or considering end offsets (**Right**).

In some cases, such as in Korea, the start and end nodes are aligned with two end sides of a road segment instead of intersection points (right side of Figure 5). With this, it is apparent that the segment is cut off by half of the length of connected roads on both sides, shortening the total actual segment distance by twice of this *end_offset* value. Since 10 m shows the highest accuracy of geocoding among 5, 10, and 15 m (not presented here), this study used this value for the *end_offset*. From the address database, the values for the $X$ and $Y$ coordinates for the start and end of the road segments, $x_{from}$, $y_{from}$, $x_{to}$, $y_{to}$, were obtained, and these values were used for the bearing angle $\theta o$. Using the *end offset* value and $\theta o$, we calculated the $X$ and $Y$ coordinates, $x_{start}$, $y_{start}$, $x_{end}$, $y_{end}$, of the shortened segment, after cut-off. Since the segment cut-off only signifies the width of the road perpendicular to the road segment in concern, the $x_{start}$ and $y_{start}$ corresponding to the beginning of the shortened segment, is still not in the middle of the sub-segment corresponding to the first building number, for example, '101' in Figure 5 on the right side. Hence, after interpolating for the location of building number '103' along

the segment, the midpoint was calculated using the quantity *mid_offset*. This midpoint is the position on the segment directly in front of the building's center.

In both types of addresses, the geocoded point's final location (*x*, *y*) is inside a building or a parcel. This final point is calculated along a pre-defined perpendicular offset, *perp offset* in Algorithm 4, away from the road. The offset can be a fixed value, like 5 m, 10 m, or a combination of different values considering road types, having an azimuth Az with a value 90° away from θo clockwise, if the building is on the right, or counterclockwise if the building is on the left. In this study, the offset is a combination of different values considering road types. Since the width of the road can vary according to different road types, we expect the combination offset to work well to move addresses on a building or a parcel correctly. Within the context of roads in Korea, the wide road type has a width of 30 m. Similarly, we assigned a width of the narrow roads with less than four lanes to 6 m, and the others to 20 m. Figure 6 illustrates this process and the procedure that follows details this.

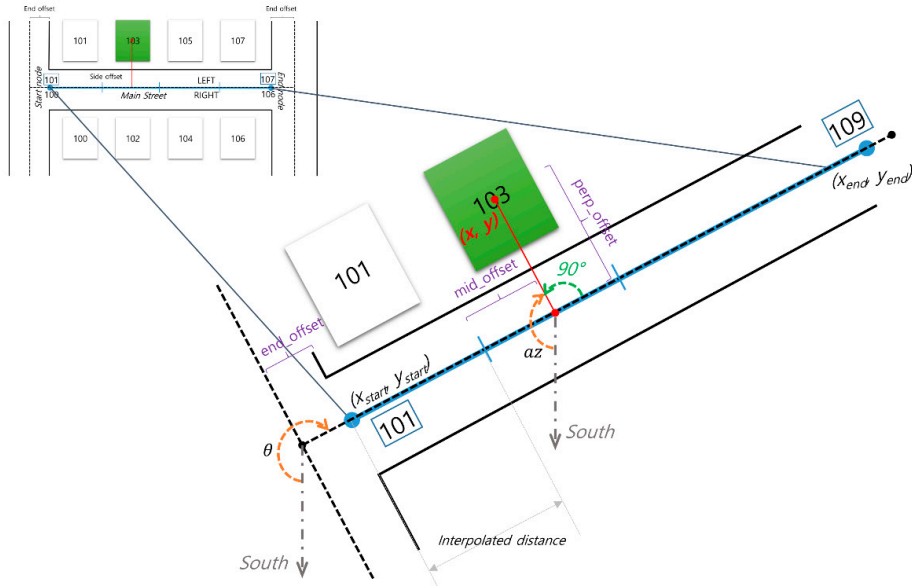

**Figure 6.** Locating the coordinates of the geocoded address.

---

**Algorithm 4** Choosing optimal thresholds for similarity metrics

---

1: **Input**: *num, addressDB, perp_offset, mid_offset, end_offset,*
2: **Output**: *x, y*
3: **Algorithm**
4: Initialize the parameters of the *direc,* $\theta_0$, *interpolate* and *Az*
5: **if** *num* is ODD
6:     **from** *addressDB* get left side *MinRange, MaxRange*
7:     *direc* = 1
8: **else**
9:     **from** *addressDB* get right side *MinRange, MaxRange*
10:     *direc* = 0
11: **from** *addressDB* get $X_{to}, X_{from}, Y_{to}, Y_{from}$
12: $\theta_0$ = calculateSouth Azimuth ($X_{to}, X_{from}, Y_{to}, Y_{from}$)
13: $X_{start} = X_{from} - end\_offset * \sin(\theta_0)$
14: $Y_{start} = Y_{from} - end\_offset * \cos(\theta_0)$
15: *interpolate* = (*num-MinRange*)/(*MaxRange-MinRange*)
16: $Az = \theta_0 - 90$
17: $x = X_{start} + (p) * (X_{end} - X_{start}) - mid\_offset * \sin(Az) - direc * perp\_offset * \sin(Az)$
18: $y = Y_{start} + (p) * (Y_{end} - Y_{start}) - mid\_offset * \cos(Az) - direc * perp\_offset * \cos(Az)$

---

## 4. Experimental Evaluation

This section presents three different kinds of experiments, as illustrated in Figure 7, to compare the performance of geocoding using machine learning with simple matching and similarity metrics. The input address data commonly go through the address parsing, matching, and locating processes. Experimental case 1 presents address matching with simple matching. This case does not use similarity metrics or machine learning for the address matching, but only simple matching described in Section 3.2. Experimental case 2 uses a similarity metric for the address matching. This case tests one similarity metric with the highest accuracy and another with the lowest accuracy in edit-based and token-based similarity metrics, respectively, to explore the ranges of performance. The Experiment case 3 uses the address matching with machine learning. For machine learning, we generated training data from the Alias DB.

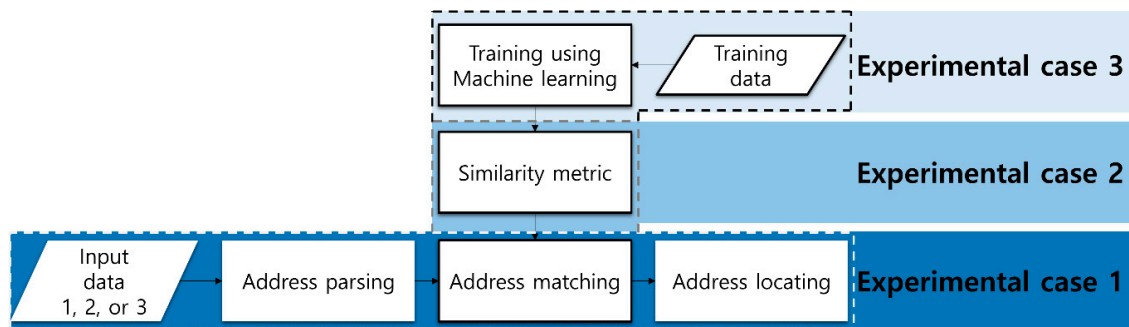

**Figure 7.** Three different kinds of experiments in this study.

For the input data, 1524 input addresses were selected and used as test data. In the input data, we assumed that city and district names were correctly input, but the road names were input with spelling errors and mistakes. In most cases, there are 25%–75% of perfect matches in input data [8], and this study generates and tests three kinds of input data with different percentages of perfect matches. Input data 1 has 30% of correct matches, Input data 2 has 50% of correct matches, and Input data 3 has 70% of correct matches. For the correct matches, we used the standard addresses obtained on the official address website and made Input data 1, 2, and 3 by changing the number of the correct addresses. To make incorrect matches in Input data 1, 2, and 3, we made wrong addresses by randomly inserting a space, special character, number, or typo, or removing a character(s) in addresses.

Figure 8 illustrates the road network data (address database in Figure 2) and its schema. It consists of road segments in Gyeonggi Province, Suwon City, Paldal District, Korea. The database schema for the road network contains the road name (*roadname*) and building numbers for address ranges. *From R* is a start building number, and *To R* is an end building number for the right side of the road. *From L* is also a start building number, and *To L* is an end building number for the left side of the road. *From X*, *From Y*, *To X*, and *To Y* are coordinates of both ends of the road segment.

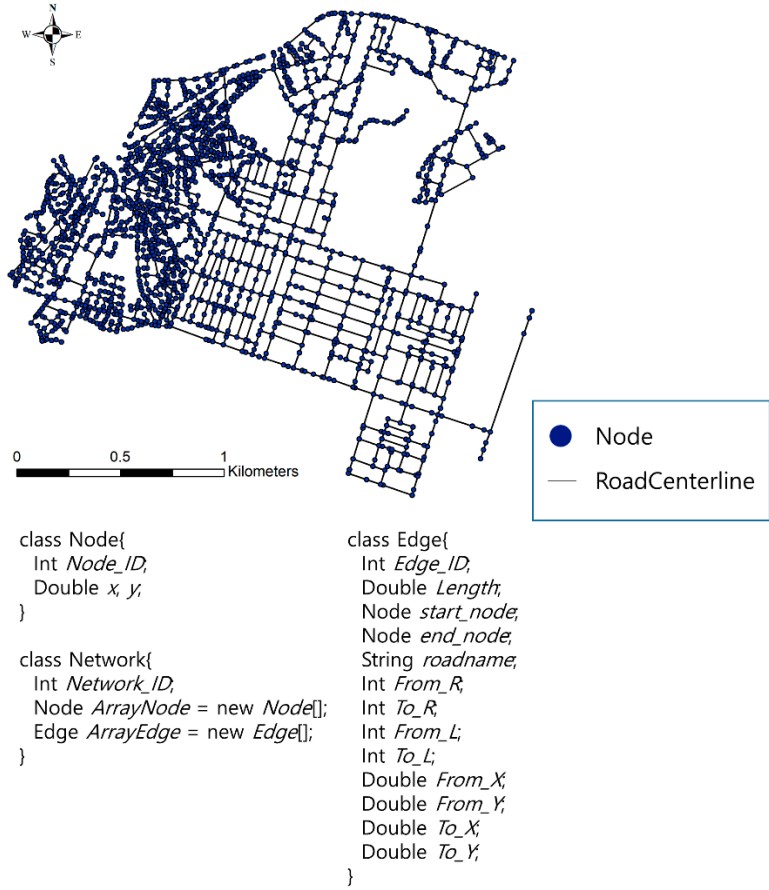

**Figure 8.** Road network data and its schema.

### 4.1. Experimental Case 1: Address Matching without Any Similarity Metrics

The input data go through the address parsing process first. The parsing process divides the input address data into city, district, road, and building number. Since we assume that there are no errors or mistakes in the city and district names and building numbers, this methodology simply matches them without a similarity metric for further processes.

The address matching without any similarity metrics shows 71.78% of accuracy when Input data 3 with 70% of correct matches are used (Table 2). As the percentage of correct matches in the input data increases, the accuracy also rises as expected. Mismatched addresses are addresses matched with incorrect addresses, whereas no addresses have matched with the unmatched addresses. Since matching for aliases with official names in the simple matching is impossible without any similarity metrics, there is no mismatched input address. Instead, there is 28.22% of unmatched input addresses in Input data 3. After that, the address locating process geocodes all the 1524 input addresses, as illustrated in Figure 9. A total of 32.74% of input addresses are geocoded on corresponding parcels using Input data 3. The same process geocoded the rest of the addresses outside the corresponding parcels.

In order to understand positional accuracy, the distances between all input addresses and corresponding parcels were also calculated and explored, excluding mismatched addresses, which may have considerable distances. On average, the geocoded points are 4.43 m further away from their corresponding parcels using Input data 3. As the percentage of correct matches increases from Input data 1 to Input data 3, the mean distance decreases because the number of addresses geocoded on corresponding parcels, which has 0 m distance, rises.

**Table 2.** Performance of address matching process without any similarity metrics, percentage of geocoded input addresses on corresponding parcels, and distances between geocoded addresses and the corresponding parcels.

| | Matched Input Address | Mismatched Input Address | Unmatched Input Address |
|---|---|---|---|
| Input data 1 * | 34.12% | | 65.88% |
| Input data 2 * | 52.82% | 0% | 47.18% |
| Input data 3 * | 71.78% | | 28.22% |
| | **Geocoded on Corresponding Parcels** | **Geocoded Outside Corresponding Parcels** | **Mean Distance ± Standard Deviation** |
| Input data 1 * | 15.09% | 84.91% | 4.53 ± 7.12 m |
| Input data 2 * | 25.20% | 74.80% | 4.45 ± 7.73 m |
| Input data 3 * | 32.74% | 67.26% | 4.43 ± 7.43 m |

* Input data 1: 30% of correct matches; Input data 2: 50% of correct matches; Input data 3: 70% of correct matches.

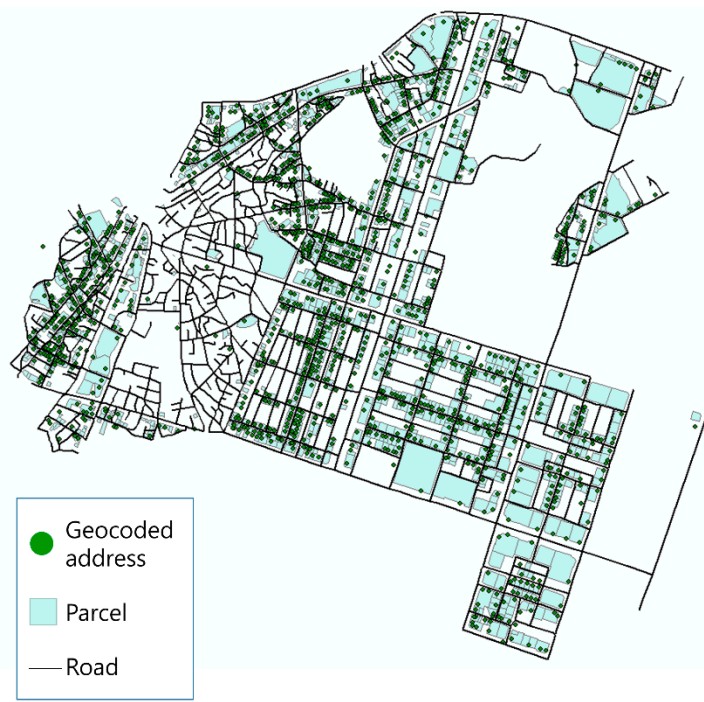

**Figure 9.** Geocoded 1524 input addresses in Gyeonggi Province, Suwon City, Paldal District, Korea.

## 4.2. Experimental Case 2: Address Matching with a Similarity Metric

In the address matching with a similarity metric, we tested Jaro-Winkler Sorted and Mlipns in edit-based similarity metrics, and Tversky and Monge-Elkan in token-based similarity metrics. Among the four metrics, Jaro-Winkler Sorted and Tversky, which have the highest accuracy when training data are used (Table 5), also achieve higher accuracy over 89% using Input data 3 than the other two metrics. Mlipns and Monge-Elkan have approximately 14% lower accuracy and 4–5 times more mismatches than Jaro-Winkler Sorted and Tversky, as shown in Table 3. The accuracy of Mlipns and Monge-Elkan becomes much lower than Jaro-Winkler Sorted and Tversky in Input data 1 and 2 than Input data 3. Regarding the quality of matches, Monge-Elkan has higher mismatched addresses than Mlipns, which means that it tries to match similar incorrect names more times. In total, 38.71% of all the input addresses are present on the corresponding parcels with the side offset in Jaro-Winkler Sorted with 70% of correct matches in input addresses, and the rest of addresses (61.29%) are outside corresponding parcels.

**Table 3.** Performance of address matching process with a similarity metric, percentage of geocoded input addresses on corresponding parcels, and distances between geocoded addresses and the corresponding parcels.

| | | Matched Input Address | Mismatched Input Address | Unmatched Input Address |
|---|---|---|---|---|
| Jaro-Winkler Sorted | | 81.23% | 8.66% | 10.11% |
| Mlipns | Input data 1* | 40.55% | 39.50% | 19.95% |
| Tversky | | 80.84% | 8.53% | 10.63% |
| Monge-Elkan | | 42.71% | 45.41% | 11.88% |
| Jaro-Winkler Sorted | | 85.30% | 6.76% | 7.94% |
| Mlipns | Input data 2* | 57.48% | 27.17% | 15.35% |
| Tversky | | 84.91% | 6.63% | 8.46% |
| Monge-Elkan | | 58.86% | 33.60% | 7.55% |
| Jaro-Winkler Sorted | | 89.76% | 4.27% | 5.97% |
| Mlipns | Input data 3* | 74.74% | 15.09% | 10.17% |
| Tversky | | 89.37% | 4.13% | 6.50% |
| Monge-Elkan | | 75.79% | 21.26% | 2.95% |
| | | **Geocoded on Corresponding Parcels** | **Geocoded Outside Corresponding Parcels** | **Mean Distance ± Standard Deviation** |
| Jaro-Winkler Sorted | | 34.51% | 65.49% | 4.40 ± 7.06 m |
| Mlipns | Input data 1* | 17.65% | 82.35% | 4.50 ± 6.95 m |
| Tversky | | 34.32% | 65.68% | 4.39 ± 7.02 m |
| Monge-Elkan | | 18.11% | 81.89% | 5.12 ± 8.03 m |
| Jaro-Winkler Sorted | | 37.01% | 62.99% | 4.42 ± 7.25 m |
| Mlipns | Input data 2* | 26.71% | 73.29% | 4.49 ± 7.63 m |
| Tversky | | 36.81% | 63.19% | 4.41 ± 7.20 m |
| Monge-Elkan | | 26.71% | 73.29% | 4.84 ± 8.07 m |
| Jaro-Winkler Sorted | | 38.71% | 61.29% | 4.45 ± 7.26 m |
| Mlipns | Input data 3* | 33.46% | 66.54% | 4.47 ± 7.40 m |
| Tversky | | 38.52% | 61.48% | 4.44 ± 7.22 m |
| Monge-Elkan | | 33.46% | 66.54% | 4.62 ± 7.57 m |

* Input data 1: 30% of correct matches; Input data 2: 50% of correct matches; Input data 3: 70% of correct matches.

The mean distances of Jaro-Winkler Sorted and Tversky are around 4.40 m, which is lower than Mlipns and Monge-Elkan when using Input data 1. Tversky has the lowest mean distance, whereas Monge-Elkan shows the highest mean distance in all the three kinds of input data.

*4.3. Experimental Case 3: Address Matching Using Machine Learning*

For the address matching using machine learning, the Alias DB and training data derived from the Alias DB are needed. The Alias DB has one official name and its four aliases. We established four aliases systematically based on the rules of making alias attributes [18], as shown in Table 4. For instance, in Alias 1, a space is inserted after the first two characters. For Alias 2, a similar character replaces the first or second character. The Alias DB has 352 records for the set of official road names and their four aliases.

**Table 4.** Four attributes of address aliases.

| | Attribute |
|---|---|
| Alias1 | Case using one space in official name |
| Alias2 | Case where the official name has only one character removed |
| Alias3 | Case where the official name has two characters removed |
| Alias4 | Case where the official name has only one misspelling |

Training data consisted of matching pairs and non-matching pairs, as described in Section 3.2 and had 1750 records. For matching pairs, each official name is paired with itself and each alias to calculate similarity values using seventeen similarity metrics. Additionally, we paired an official name and its four aliases with another similar official name, to make non-matching pairs. The reason why we chose similar names for non-matching pairs is to increase the performance of machine learning models when distinguishing a name against its similar form. One example of the non-matching

pairs is <Hyowon 94th street, Hyowon 9th street>. We generated and tested three kinds of training data consisting of 20% of matching pairs and 80% of non-matching pairs (Training data 1), 50% of matching pairs and 50% of non-matching pairs (Training data 2), or 80% of matching pairs and 20% of non-matching pairs (Training data 3). The training data have a column 'Matching' used as a label for training, which provides information on whether the records are the matching pairs (Matching: 1) or non-matching pairs (Matching: 0).

In the address matching, to choose the best machine learning model, the performance of three machine learning models was compared with the matching without any similarity metrics and 17 similarity metrics using training data through the model evaluation and selection. Without any similarity metrics, matching shows 85% of accuracy using Training data 1 and 37% of accuracy using Training data 3 (Table 5). As the ratio of matching pairs is becoming more substantial, the matching accuracy without similarity metrics becomes smaller too because the matching only identifies the same word pairs, and this method cannot regard similar words as identical. Among different similarity metrics, Overlap shows the highest accuracy using Training data 1 (95%), whereas Jaro-Winkler Sorted, Tversky, and Jaccard show the best accuracy using Training data 3 (96%). Most of similarity metrics show the highest accuracy in Training data 3 when compared to Training data 1 and 2, which is opposite to the matching without similarity metrics, possibly because the similarity metrics can match similar words, and the Training data 3 has the highest percentage of matching pairs with aliases. However, when the training data are 50% + 50% pairs and are not biased to the matching or non-matching pairs, many similarity metrics have the lowest accuracy. It indicates that it is difficult for similarity metrics to successfully identify both matching and non-matching pairs well when the matching and non-matching pairs are of equal parts in the training data.

**Table 5.** Performance of address matching without any similarity metrics, 17 similarity metrics, and three machine learning methods.

| Method | Training Data 1 * | Training Data 2 * | Training Data 3 * |
|---|---|---|---|
| Matching without similarity metrics | 85 | 61 | 37 |
| Jaro | 89 | 92 | 95 |
| Jaro-Winkler | 89 | 85 | 92 |
| Jaro-Winkler Reversed | 92 | 90 | 95 |
| Jaro-Winkler Sorted | 91 | 91 | 96 |
| Hamming | 85 | 67 | 80 |
| Mlipns | 58 | 59 | 80 |
| StrCmp95 | 89 | 87 | 94 |
| Needleman_Wunsch | 85 | 68 | 84 |
| Gotoh | 85 | 81 | 90 |
| Smith_Waterman | 85 | 76 | 87 |
| Cosine | 89 | 89 | 94 |
| Tversky | 90 | 91 | 96 |
| Overlap | 95 | 89 | 92 |
| Bag | 85 | 80 | 90 |
| Jaccard | 90 | 91 | 96 |
| Sorensen_Dice | 89 | 89 | 94 |
| Monge-Elkan | 87 | 63 | 80 |
| SVM | 100 | 99 | 97 |
| RF | 100 | 99 | 98 |
| XGB | 100 | 100 | 98 |

* Training data 1: 20% of matching pairs and 80% of non-matching pairs; Training data 2: 50% of matching pairs and 50% of non-matching pairs; Training data 3: 80% of matching pairs and 20% of non-matching pairs.

Three machine learning models have higher accuracy than all the 17 similarity metrics in the three kinds of training data with different pair combinations. All the three methods are tuned with optimal hyper-parameters to achieve the best performance. We performed hyper-parameter tuning

using a grid search. This method sets a list of parameters, and the range of values for each parameter. For each classifier, the algorithm attempts every combination of parameters. Then, the best set of hyper-parameter values was chosen based on accuracy. SVM, RF, and XGB have the highest accuracy using Training data 1 (100%), and it gets a bit lower as the ratio of matching pairs increases. SVM and RF show 99% of accuracy using Training data 2, while XGB achieves 100%. Using Training data 3, SVM has 97% of accuracy, whereas RF and XGB show 98%. XGB shows the highest accuracy (100%) in Training data 2, tuned with 0.05 for learning rate, 400 gradient boosted trees, and 0.7 for subsample ratio of columns.

Among the three kinds of training data, Training data 3, which shows the lowest accuracy of machine learning models, was used to test different numbers of similarity metrics and select the best machine learning model. We chose 97% as the minimum accuracy that all the three machine learning models need to exceed, which is 1% higher than the highest accuracy of similarity metrics in Training data 3. We determined different kinds of similarity metrics tested by descending order of their accuracy. As shown in Figure 10, all the trained models exceed 97% using nine similarity metrics (Jaro-Winkler Sorted, Tversky, Jaccard, Jaro, Jaro-Winkler Reversed, Hamming, Cosine, Sorensen_Dice, Jaro-Winkler). Among them, XGB shows the highest accuracy (98.29%) and is the best model with nine similarity metrics for the evaluation of geocoding.

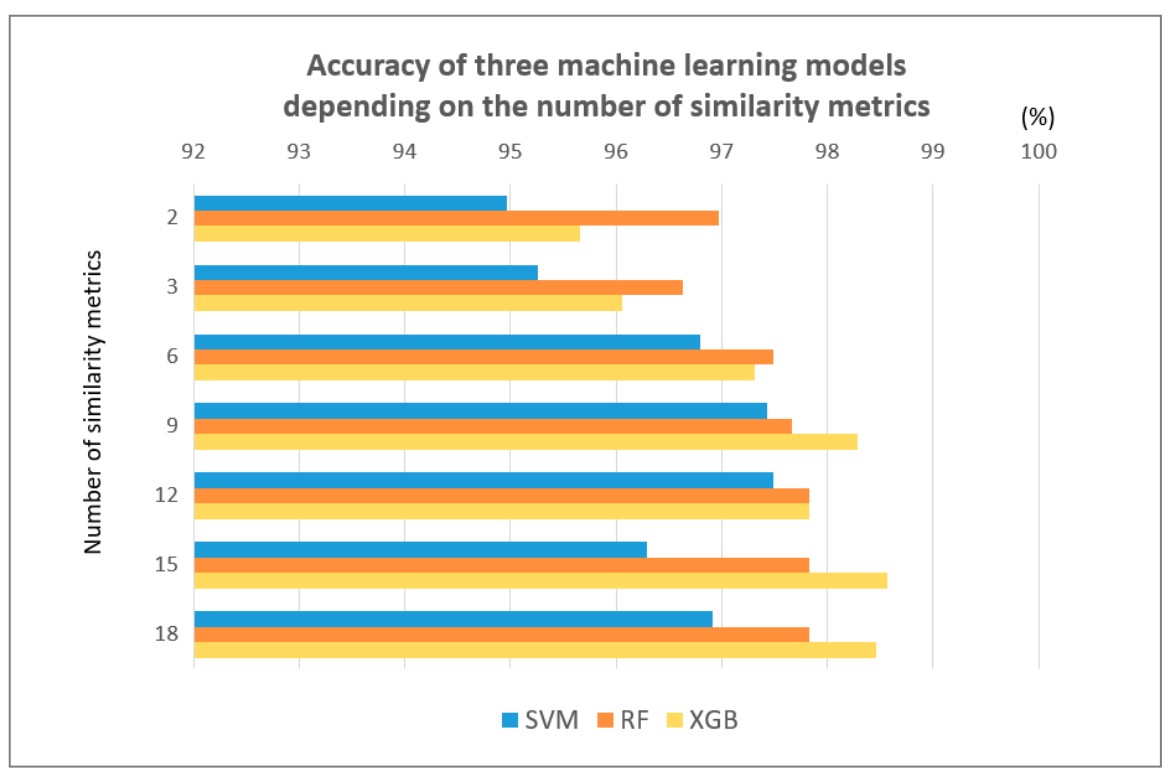

**Figure 10.** Accuracy of three machine learning models depending on the number of similarity metrics.

With the trained XGB model, as a result, 96.39% of input addresses are correctly matched through the address matching process using machine learning (Table 6) in Input data 3. Since some input addresses include random spelling errors and mistakes that do not abide by the systematic rules, the accuracy becomes lower than the results in Table 5. Regarding the quality of address matching, there is 1.25% of mismatched addresses and 2.36% of unmatched addresses in Input data 3. Results show that 40.94% of all the input addresses are geocoded on the corresponding parcels with the side offset, and the rest of the addresses (59.06%) are outside corresponding parcels. The percentage of the addresses geocoded on corresponding parcels is not very different in three kinds of input data with different ratios of correct matches.

**Table 6.** Performance of address matching process with a similarity metric, percentage of geocoded input addresses on corresponding parcels, and distances between geocoded addresses and the corresponding parcels.

|  | **Matched Input Address** | **Mismatched Input Address** | **Unmatched Input Address** |
|---|---|---|---|
| Input data 1 * | 95.74% | 1.90% | 2.36% |
| Input data 2 * | 96.33% | 1.31% | 2.36% |
| Input data 3 * | 96.39% | 1.25% | 2.36% |
|  | **Geocoded on Corresponding Parcels** | **Geocoded Outside Corresponding Parcels** | **Mean Distance ± Standard Deviation** |
| Input data 1 * | 40.62% | 59.38% | 4.44 ± 7.09 m |
| Input data 2 * | 40.94% | 59.06% | 4.48 ± 7.21 m |
| Input data 3 * | 40.94% | 59.06% | 4.47 ± 7.21 m |

* Input data 1: 30% of correct matches; Input data 2: 50% of correct matches; Input data 3: 70% of correct matches.

On average, the geocoded points are 4.44 m further away from their corresponding parcels in Input data 1. The mean distance of Input data 1 is lower than the other two kinds of input data because a few addresses geocoded with long distances are unmatched, so the evaluation excludes the distances of these addresses. The 4 m distance on average with 7 m standard deviation is similar to GPS positional accuracy. GPS positional accuracy ranges from 4.4 to 10.3 m under an urban environment [48], so considering this, the developed geocoding system is applicable in some domains, like GPS-equipped car navigation.

*4.4. Comparison of the Results of Three Experimental Cases*

Table 7 and Figure 11 show the results of all the three experimental cases. The accuracy of the address matching with or without similarity metrics and machine learning increases along with the ratio of correct matches, which increases from Input data 1 to Input data 3. Results show that address matching using machine learning outperforms the matching with or without similarity metrics. Notably, the address matching using XGB is 7% higher than Jaro-Winkler Sorted and Tversky and 21% higher than Mlipns and Monge-Elkan when using input data with 70% of correct matches. When input data have 30% of correct matches (Input data 1), the difference of accuracy between XGB and Jaro-Winkler Sorted and Tversky becomes 14%, which is two times more than input data with 70% of correct matches. Further, the difference of accuracy between XGB and Mlipns and Monge-Elkan becomes 53–55%, which is 2.5 times more than input data having 70% of correct matches. It indicates that the performance of Mlipns and Monge-Elkan drops more than Jaro-Winkler Sorted and Tversky as the percentage of correct matches in the input data decreases.

**Table 7.** Accuracy of address matching in three experimental cases.

| **Experimental Case** |  | **Input Data 1 *** | **Input Data 2 *** | **Input Data 3 *** |
|---|---|---|---|---|
| Experimental case 1 |  | 34.12% | 52.82% | 71.78% |
| Experimental case 2 | Jaro-Winkler Sorted | 81.23% | 85.30% | 89.76% |
|  | Mlipns | 40.55% | 57.48% | 74.74% |
|  | Tversky | 80.84% | 84.91% | 89.37% |
|  | Monge-Elkan | 42.71% | 58.86% | 75.79% |
| Experimental case 3 | XGB | 95.74% | 96.33% | 96.39% |

* Input data 1: 30% of correct matches; Input data 2: 50% of correct matches; Input data 3: 70% of correct matches.

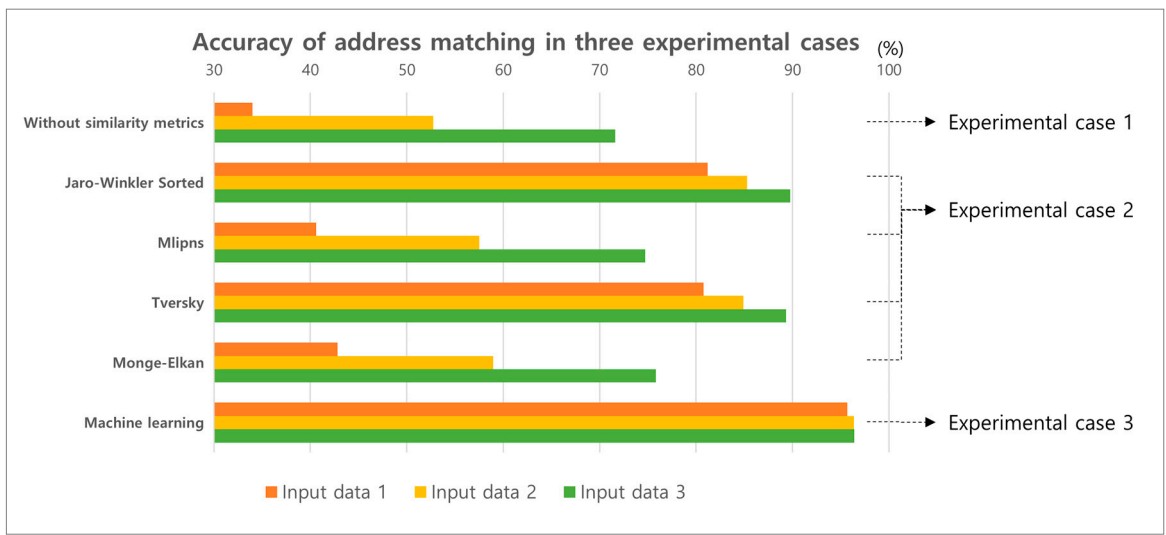

**Figure 11.** Accuracy of address matching in three experimental cases.

The performance of XGB is consistently high when compared to address matching with or without similarity metrics. The accuracy of XGB is 95–96% across input data with different percentages of correct matches, whereas the accuracy of address matching with or without similarity metrics is relatively low and varies more. Especially, Jaro-Winkler Sorted and Tversky vary from around 80% in input data with 30% of correct matches, to 89% in input data with 70% of correct matches. When XGB gains 1%, Jaro-Winkler Sorted and Tversky change 9%. As a result, the address matching using machine learning is more stable and accurate to use for geocoding than simple matching or matching with a similarity metric. Combining the capabilities of various similarity metrics using machine learning is better than using each similarity metric for geocoding.

Further, among the four kinds of similarity metrics, Jaro-Winkler Sorted and Tversky are more consistent across input data with different ratios of correct matches than Mlipns and Monge-Elkan. When Jaro-Winkler Sorted and Tversky vary 9%, Mlipns and Monge-Elkan change approximately 34%. Jaro-Winkler Sorted and Tversky, which have the highest accuracy when using training data, also achieve high accuracy in input data and are found to be less sensitive to the proportion of correct matches in address matching. On the other hand, Mlipns and Monge-Elkan in edit-based and token-based similarity metrics, which have the lowest accuracy when using training data, also show low accuracy and are found to be more sensitive to the proportion of correct matches. Thus, using Mlipns and Monge-Elkan is not expected to show consistent performance and not appropriate for accurate address matching when there are input addresses with different percentages of correct matches.

## 5. Conclusions

This study suggested an algorithm for accurate and stable geocoding. Address parsing helped to obtain meaningful units of addresses to send them to the matching process. We introduced machine learning techniques in order to achieve high accuracy in the address matching process. It was proved that XGB with the optimal hyper-parameters was the best machine learning method with the highest accuracy in the address matching process, and the accuracy of XGB outperformed similarity metrics when using training data or input data. The performance of XGB was also consistent across different kinds of input data. The address matching process using machine learning was able to deal with human errors, including spelling errors, in input addresses to match addresses accurately. As a module in the suggested geocoding system, it can be applied to any other geocoding system to precisely convert addresses into geographic coordinates for relevant research and applications. The address locating allowed to narrow down to a road segment from the candidate road segments selected in the address matching and convert addresses into one geocoded point on a map.

This study, however, has some limitations. First, the performance of the matching process was dependent on the quality of the Alias DB. Training data only had 1750 records for matching and non-matching pairs, which is an insufficient number to make a robust machine learning method. For each official name, we only made four different kinds of aliases. There may be some excluded aliases in the Alias DB and training process. Thus, for future work, more matching cases with more aliases need to be considered in machine learning methods to enhance the robustness of the trained methods. Second, the proposed algorithm was only applicable for street-based addresses. Apart from the street-based addresses, there are area-based addressing systems and hybrid addressing systems [5], and the proposed algorithm did not consider those two addressing systems. Hierarchical area-based addressing systems are used in Eastern Asia, while China has a hybrid addressing system. In order to apply to various countries, we suggest the expansion of the proposed algorithm to embody the two different addressing systems for future research.

Moreover, we need to put some efforts on decreasing mismatched addresses. Mismatched addresses ended up being geocoded in other locations far away from correct places and decreased the positional accuracy of geocoding. Therefore, by finding a way to complement the machine learning (e.g., adding additional rules), we need to avoid making mismatching. Lastly, this study did not consider unstructured addresses in the parsing step of the geocoding. This study assumed that the input addresses are structured and normalized, and no error occurred in the parsing process. Although machine learning techniques in the matching process can handle abbreviations, misspellings, and misplacements, to achieve high performance of the geocoding, the normalization, like removing punctuation and standardization is necessary to some extent [49]. Therefore, future work needs to strengthen the parsing process.

**Author Contributions:** Conceptualization, J.L. and K.L.; methodology, K.L. and A.R.C.C.; software, K.L. and A.R.C.C.; validation, K.L., A.R.C.C. and J.L.; formal analysis, K.L. and A.R.C.C.; investigation, K.L.; resources, K.L.; data curation, K.L.; writing—original draft preparation, K.L. and A.R.C.C.; writing—review and editing, A.R.C.C. and J.L.; visualization, K.L.; supervision, J.L.; project administration, J.L.; funding acquisition, J.L. All authors have read and agreed to the published version of the manuscript.

**Funding:** This research was supported by Basic Science Research Program through the National Research Foundation of Korea (NRF) funded by the Ministry of Education (No. 2017R1D1A1B03028890).

**Conflicts of Interest:** The authors declare no conflict of interest.

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
