# Peer review of "Improving a Street-Based Geocoding Algorithm Using Machine Learning Techniques"

_applsci, doi:10.3390/app10165628_

Round 1

Reviewer 1 Report

The paper "A machine learning approach for street-based address geocoding" focuses on the testing of the three different machine learning methods for street based address geocoding: SVM, RF, eXtreme Gradient Boosting (XGB). In this study there are introduced the similarity metrics for better performance of matching by applying different ratio of matching and non-matching input address. The input address data commonly go through the address parsing, matching, and locating processes. Three experimental cases have been introduced by the authors to compare the performance of geocoding using machine learning with other approaches, namely: experimental case 1 presented address matching with simple matching; experimental case 2 used a similarity metric for the address matching, while the experiment case 3 used the address matching  with machine learning. It has been found that the performance of XGB was consistently high when compared to address matching with or without similarity metrics. The accuracy of XGB was ranging from 95% to 96% across input data with different percentages of correct matches, whereas the accuracy of address matching with or without similarity metrics was relatively low and varies more.

Usually, SVM has an accuracy of 97% and RF an accuracy of 98%. Please insert in the paper the results obtained for this methods in order to compare with XGB and to prove your conclusions.

Author Response

Cover Letter and Responses to Comments

Improving a Street-based Geocoding Algorithm using Machine Learning Techniques

05 August 2020

CRISTIANO MATRICARDI, PhD

Editor

Applied Sciences

Dear Dr. Matricardi and the reviewers,

We are submitting our revised manuscript for a research article entitled Improving a Street-based Geocoding Algorithm using Machine Learning Techniques with Manuscript-ID applsci-889111 for consideration of publication in the Special Issue of the Applied Sciences Journal, entitled Remote Sensing and Geoscience Information Systems in Applied Sciences. The aforementioned article was revised from its original title A Machine Learning Approach for Street-Based Address Geocoding in the previous version to better reflect the work we have described therein.

We sincerely thank the reviewers for your provided feedback, and we have incorporated changes that reflect most of your suggestions. We have provided these point-by-point responses below and highlighted the corresponding changes within the revised manuscript which you can see through the Track Changes function in Microsoft Word.

Similar to our previous revision, we declare that no part of this original manuscript has been published elsewhere, and it has not been submitted simultaneously in another journal. Furthermore, we declare no conflicts of interest in this work. We are thankful for this consideration.

Yours Sincerely,

The Authors

Reviewer #1

  1. The paper "A machine learning approach for street-based address geocoding" focuses on the testing of the three different machine learning methods for street based address geocoding: SVM, RF, eXtreme Gradient Boosting (XGB). In this study there are introduced the similarity metrics for better performance of matching by applying different ratio of matching and non-matching input address. The input address data commonly go through the address parsing, matching, and locating processes. Three experimental cases have been introduced by the authors to compare the performance of geocoding using machine learning with other approaches, namely: experimental case 1 presented address matching with simple matching; experimental case 2 used a similarity metric for the address matching, while the experiment case 3 used the address matching with machine learning. It has been found that the performance of XGB was consistently high when compared to address matching with or without similarity metrics. The accuracy of XGB was ranging from 95% to 96% across input data with different percentages of correct matches, whereas the accuracy of address matching with or without similarity metrics was relatively low and varies more.

Author response: Thank you for your support for our research and your valuable comments. We appreciate the time and effort you have dedicated to providing feedback on our manuscript for our improvement. In this research, we explored how machine learning methods would assist geocoding in the inevitable presence of human errors by developing a module to match incorrect addresses to produce accurate point locations on a map. This module can be useful for applications that require converting addresses to geographic coordinates.

  1. Usually, SVM has an accuracy of 97% and RF an accuracy of 98%. Please insert in the paper the results obtained for this methods in order to compare with XGB and to prove your conclusions.

Author response: Thank you for this comment. We have added the results of SVM and RF on the Training data 3 (97% and 98%). The added sentence is “SVM and RF show 99% accuracy using Training data 2, while XGB achieves 100%. Using Training data 3, SVM has 97% accuracy, whereas RF and XGB show 98%.” in Section 4.3 (page 18, line 434-435).

Reviewer 2 Report

This study employs machine learning models in the address matching process to obtain high accuracy. This paper is well written, and the reviewer has some minor comments.

  • The tables are not numbered consecutively.
  • Explain the way to tune hyper-parameters of the machine learning models.
  • How do the authors prepare input data 1, 2, and 3 of Chapter 4?

Author Response

Cover Letter and Responses to Comments

Improving a Street-based Geocoding Algorithm using Machine Learning Techniques

05 August 2020

CRISTIANO MATRICARDI, PhD

Editor

Applied Sciences

Dear Dr. Matricardi and the reviewers,

We are submitting our revised manuscript for a research article entitled Improving a Street-based Geocoding Algorithm using Machine Learning Techniques with Manuscript-ID applsci-889111 for consideration of publication in the Special Issue of the Applied Sciences Journal, entitled Remote Sensing and Geoscience Information Systems in Applied Sciences. The aforementioned article was revised from its original title A Machine Learning Approach for Street-Based Address Geocoding in the previous version to better reflect the work we have described therein.

We sincerely thank the reviewers for your provided feedback, and we have incorporated changes that reflect most of your suggestions. We have provided these point-by-point responses below and highlighted the corresponding changes within the revised manuscript which you can see through the Track Changes function in Microsoft Word.

Similar to our previous revision, we declare that no part of this original manuscript has been published elsewhere, and it has not been submitted simultaneously in another journal. Furthermore, we declare no conflicts of interest in this work. We are thankful for this consideration.

Yours Sincerely,

The Authors

Reviewer #2

  1. This study employs machine learning models in the address matching process to obtain high accuracy. This paper is well written, and the reviewer has some minor comments. 

Author response: We sincerely thank you for the valuable time and effort you devoted to reviewing and providing feedback towards the improvement of our manuscript. We address your minor comments in the items below.

  1. The tables are not numbered consecutively. 

Author response: Thank you for pointing this mistake out. We have corrected the table numbers accordingly. (page 6, line 198; page 14, line 363; page 15, line 386; page 18, line 438; page 19, line 472)

  1. Explain the way to tune hyper-parameters of the machine learning models. 

Author response: Thank you for pointing out this important question. We have added the explanation of grid search that we used for hyper-parameter tuning. The added explanation is “We perform hyper-parameter tuning using a grid search. This method sets a list of parameters, and the range of values for each parameter. For each classifier, the algorithm attempts every combination of parameters. Then, the best set of hyper-parameter values is chosen based on accuracy.” in Section 4.3 (page 17, line 429-432).

  1. How do the authors prepare input data 1, 2, and 3 of Chapter 4?

Author response: Thank you for your comment. We understand that our discussion regarding the preparation of the input data for our experiments needs improvement. In order to describe the process more concretely, we have added discussions on how the input data are made with correct and incorrect matches. The added discussions are “For the correct matches, we used the standard addresses obtained on the official address website and made Input data 1,2, and 3 by changing the number of the correct addresses. To make incorrect matches in Input data 1,2, and 3, we made wrong addresses by randomly inserting a space, special character, number, or typo or removing a character(s) into addresses.” in Section 4 (page 12, line 328-332).

Other revisions

  1. We have revised the title from “A Machine Learning Approach for Street-Based Address Geocoding” to “Improving a Street-based Geocoding Algorithm using Machine Learning Techniques” to more accurately describe our work presented in the paper. (page 1, line 2-3)

  1. We have also replaced Figure 1 with a figure of better quality. (page 3, line 101)

  1. Minor punctuation and grammatical error fixes at the following locations.
    1. page 1, line 21, 24, 35
    2. page 2, line 90
    3. page 5, line 188
    4. page 15, line 367
    5. page 21, line 522
    6. page 22, line 542-543

Round 2

Reviewer 1 Report

The paper "Improving a Street-based Geocoding Algorithm 2 using Machine Learning Techniques" focuses on the testing of the three different machine learning methods for street based address geocoding: SVM, RF, eXtreme Gradient Boosting (XGB). In this study there are introduced the similarity metrics for better performance of matching by applying different ratio of matching and non-matching input address. The input address data commonly go through the address parsing, matching, and locating processes. Three experimental cases have been introduced by the authors to compare the performance of geocoding using machine learning with other approaches, namely: experimental case 1 presented address matching with simple matching; experimental case 2 used a similarity metric for the address matching, while the experiment case 3 used the address matching  with machine learning. It has been found that the performance of XGB was consistently high when compared to address matching with or without similarity metrics. The accuracy of XGB was ranging from 95% to 96% across input data with different percentages of correct matches, whereas the accuracy of address matching with or without similarity metrics was relatively low and varies more.